# *CCNB2* and *AURKA* overexpression may cause atypical mitosis in Japanese cortisol-producing adrenocortical carcinoma with *TP53* somatic variant

**Akira Ikeya**[1☯], **Mitsuko Nakashima**[2☯], **Miho Yamashita**[3*‡], **Keisuke Kakizawa**[1], **Yuta Okawa**[1], **Hirotomo Saitsu**[2,3], **Shigekazu Sasaki**[2], **Hironobu Sasano**[4], **Takafumi Suda**[1], **Yutaka Oki**[5‡]

**1** 2nd Department of Internal Medicine, Hamamatsu University School of Medicine, Shizuoka, Japan, **2** Department of Biochemistry, Hamamatsu University School of Medicine, Shizuoka, Japan, **3** Department Internationalization Center, Hamamatsu University School of Medicine, Shizuoka, Japan, **4** Department of Pathology, Tohoku University Graduate School of Medicine, Miyagi, Japan, **5** Department of Family and Community Medicine, Hamamatsu University School of Medicine, Shizuoka, Japan

☯ These authors contributed equally to this work.
‡These authors also contributed equally to this work.
* mihojy@hama-med.ac.jp

**Data Availability Statement:** https://www.cbioportal.org/ All relevant data are within the manuscript and its Supporting Information files.

## Abstract

### Background

Many genomic analyses of cortisol-producing adrenocortical carcinoma (ACC) have been reported, but very few have come from East Asia. The first objective of this study is to verify the genetic difference with the previous reports by analyzing targeted deep sequencing of 7 Japanese ACC cases using next-generation sequencing (NGS). The second objective is to compare the somatic variant findings identified by NGS analysis with clinical and pathological findings, aiming to acquire new knowledge about the factors that contribute to the poor prognosis of ACC and to find new targets for the treatment of ACC.

### Method

DNA was extracted from ACC tissue of seven patients and two reference blood samples. Targeted deep sequencing was performed using the MiSeq system for 12 genes, and the obtained results were analyzed using MuTect2. The hypothesis was obtained by integrating the somatic variant findings with clinical and pathological data, and it was further verified using The Cancer Genome Atlas (TCGA) dataset for ACC.

### Results

Six possible pathogenic and one uncertain significance somatic variants including a novel *PRKAR1A* (NM_002734.4):c.545C>A (p.T182K) variant were found in five of seven cases. By integrating these data with pathological findings, we hypothesized that cases with *TP53* variants were more likely to show atypical mitotic figures. Using TCGA dataset, we found

**Funding:** MN the Takeda Science Foundation and a HUSM Grant-in-Aid from Hamamatsu University School of Medicine. https://www.takeda-sci.or.jp/business/abroad_e.html.

**Competing interests:** The authors have declared that no competing interests exist.

that atypical mitotic figures were associated with *TP53* somatic variant, and mRNA expression of *CCNB2* and *AURKA* was significantly high in *TP53* mutated cases and atypical mitotic figure cases.

## Conclusion

We believe this is the first report that discusses the relationship between atypical mitotic figures and *TP53* somatic variant in ACC. We presumed that overexpression of *CCNB2* and *AURKA* mRNA may cause atypical mitosis in *TP53* somatic mutated cases. Because *AURKA* is highly expressed in atypical mitotic cases, it may be an appropriate indicator for AURKA inhibitors.

## Introduction

Adrenocortical carcinoma (ACC) is rare, with an annual incidence of 0.7–2.0 cases per million [1,2], but aggressive. The 5-year survival rate for patients with ACC is 35%, and the effects of pharmacotherapy are limited and not yet well established [3]. Application of the Weiss criteria (nuclear atypia, mitotic rate, atypical mitotic figures, clear cells ≤ 25%, diffuse architecture, necrosis, venous invasion, sinusoidal invasion, and capsular invasion) is the most commonly used method for making histopathological ACC diagnosis [4,5]. ACC can be diagnosed by the presence of at least three of the nine Weiss criteria. However, it is difficult in some cases to distinguish ACC from adrenocortical adenoma (ACA). The reproducibility of Weiss scores and in particular the inter-individual reproducibility is often a problem. Ki-67 is a proliferation index immunomarker that can help to refine the diagnosis and prognosis of ACC. It is established that ACC mostly shows a Ki-67 index ≥ 5% [6]. The Ki-67 index is also a valuable prognostic marker [7,8]. The European Network for the Study of Adrenal Tumors (ENSAT) guidelines suggest that patients with a Ki-67 index ≥ 10% should be perceived as having a high risk of recurrence [9]. Additionally, it is also known that cortisol production is involved as a prognostic predictor in ACC. The association between cortisol-producing ACC and mortality shows a positive hazard ratio of 1.71 (95% confidence interval of 1.18–2.47) after adjusting for tumor stage. Further, ENSAT guidelines define cortisol-producing ACC as a poor prognostic factor [9,10]. However, the mechanism by which patients with cortisol-producing ACC have a poor prognosis has not been elucidated.

Recently, many genomic and molecular analyses of ACC have been conducted, suggesting new prognostic factors such as specific somatic variants and hypermethylations [11,12]. In ACC, gene alterations such as *TP53* R337H germline variants of pediatric ACC [13] and *ZNRF3* copy number changes [14] have been reported as gene alterations belonging to the TP53-RB1 and Wnt pathways, respectively. In particular, somatic variants in *TP53* (16–20%) and *CTNNB1* (about 15%) have been the most frequently reported in ACC [14,15]. Somatic variants in these two genes are reported as a poor prognostic factor [12,16,17,18]. However, how the somatic variant in *TP53* or *CTNNB1* affects clinical and pathological findings and leads to poor prognosis of ACC is not fully elucidated.

The Cancer Genome Atlas (TCGA) project for ACC (ACC-TCGA) is the largest open-source database for comprehensive genomic and molecular analysis of ACC [14], and many studies use the ACC-TCGA dataset for genomic and molecular secondary analyses [19,20]. Those large-scale studies include whole genome or exome sequencing data using next-generation sequencing (NGS) technology. These genome sequencing approaches have an advantage

in exhaustive screening for entire genomic alterations but have limitations in sensitivity and specificity for detecting low-prevalence somatic variants because of the relatively low coverage of reads. Previous studies suggested that the low-prevalence somatic variants affect tumorigenesis; therefore, high-sensitivity variant screening methods are required for cancer genome analysis. Targeted deep sequencing is a useful tool for detecting low-prevalence somatic variants by analyzing a subset of the genes of interest with deep depth of coverage and improves the sensitivity and specificity of variant detection [21,22].

Additionally, genomic and molecular analysis of ACC have been conducted mostly in the United States and Europe, and very few studies of ACC have been performed using cases from East Asia, including Japan [14,15,23,24,25]. For this reason, only insufficient genomic information exists for ACC in the Japanese population. Therefore, in this study, we examined somatic variants in cortisol-producing ACC tissues from 7 different Japanese individuals by targeted sequencing for 12 candidate genes, that enable us to achieve deep coverage to detect low-prevalence somatic variants. We selected candidate genes, in which somatic variants might be related to poor prognosis. Ten genes (*APC*, *CCNE1*, *CDK4*, *CDKN2A*, *CTNNB1*, *MDM2*, *MEN1*, *RB1*, *TP53*, and *ZNRF3*) involved in TP53-RB1 and Wnt pathways were selected because *TP53* and *CTNNB1* somatic variants have been reported as poor prognostic factors in ACC [12,16,17,18]. Somatic variants in these ten candidate genes were detailed in previous papers and were suitable for a comparative study between Japanese and other races. In addition, we further selected *PRKAR1A* and *TERF2* which are both reported as driver genes of ACC. *PRKAR1A* is associated with cortisol production, which is a factor of poor prognosis. *TERF2* may be associated with disease progression due to shortened telomere length [14]. We assessed the prognosis factors of ACC by combining the results of somatic variant analysis with clinical and pathological findings, and performed additional verification using the ACC-TCGA dataset. From these analyses, we considered the possibility of new treatment strategies that could lead to personalized medicine for ACC.

## Materials and methods

### Patients and data

We analyzed the cases of seven Japanese patients with cortisol-producing ACC who were diagnosed and operated on at Hamamatsu University School of Medicine Hospital from July 1993 to February 2019. Genomic DNA and total RNA were extracted from cancer lesion tissues of all subjects (cases 1 to 7), and peripheral blood samples were drawn from two surgical subjects (cases 6 and 7) as reference. We selected the same number of cortisol-producing ACA as ACC to use as a reference for mRNA expression analysis by real-time quantitative polymerase chain reaction (PCR). Among the ACA cases in which fresh frozen tissue samples were stored, those cases that were operated on during the same periods as ACC were extracted. Among them, we excluded those with insufficient clinical information, selected 7 cases in descending order of sample volume.

Samples from ACC and ACA tissues were obtained during surgery, blood was obtained during blood tests. Blood sample was immediately subjected to DNA extraction, and tissue samples were stored at -80˚C with All protect Tissue Reagent (Qiagen, Hilden, Germany) until subjected to DNA and RNA extraction. We collected the following clinical parameters: age at diagnosis, sex, tumor size, cortisol level at 1-mg or 8-mg dexamethasone suppression test (DST), hormone secretion, pathological stage following ENSAT2008 classification, Weiss score, positive or negative result for each Weiss factor, Ki-67 index (in percent), survival status, and survival time. The study received ethical approval from the Hamamatsu University School of Medicine review boards (approval no. 17–260).

## DNA extraction and PCR amplification

Genomic DNA (gDNA) was isolated from frozen ACC tissues and blood samples using the QIAamp Fast DNA Tissue Kit and the QIAamp DNA Blood Mini Kit (Qiagen, Hilden, Germany), respectively, according to the manufacturer's protocols. gDNA samples were quality checked using a NanoDrop 1000 Spectrophotometer (Thermo Fisher, Waltham, USA), and an absorbance ratio greater than 1.8 for 260/280 nm was confirmed.

To detect the candidate somatic variants, we performed PCR deep sequencing using targeted gene-specific primers. Primers were designed for 12 candidate genes–*APC*, *CCNE1*, *CDK4*, *CDKN2A*, *CTNNB1*, *MDM2*, *MEN1*, *PRKAR1A*, *RB1*, *TERF2*, *TP53*, and *ZNRF3* –covering the coding region, intron–exon boundaries, and 5′ and 3′ untranslated regions (see S1 Table). Each 10-µl PCR solution contained 5 µl of 2 × PCR Buffer for KOD FX Neo, 2 µl of 2-mM dNTPs, 1 µl of 2-µM primer mix (0.2-µM final concentration of each primer), 0.2 µl of KOD FX Neo polymerase (TOYOBO, Osaka, Japan), 0.8 µl Nuclease-Free water (Ambion, Austin, USA) and 1 µl of DNA template (20–30 ng gDNA). PCR amplifications were used a two- or three-step touchdown PCR protocol. The two-step PCR protocol is as follows: initial denaturation (94˚C, 2 min) followed by 35 cycles at 98˚C for 10 s, 68˚C for 4 min, and a final extension step at 68˚C for 5 min. The three-step touchdown PCR protocol is as follows: initial denaturation (94˚C, 2 min) followed by 5 cycles at 98˚C for 10 s, 62˚C for 30 s, and 68˚C for 4 min; 5 cycles at 98˚C for 10 s, 60˚C for 30 s, and 68˚C for 4 min; 5 cycles at 98˚C for 10 s, 58˚C for 30 s, and 68˚C for 4 min; 25 cycles at 98˚C for 10 s, 56˚C for 30 s, and 68˚C for 4 min; and a final extension step at 68˚C for 5 min. We confirmed the product sizes of PCR amplification products by agarose gel electrophoresis.

## NGS

A dual-indexed sequencing library was prepared with the Nextera DNA Flex Library Prep Kit (Illumina, San Diego, USA) and sequenced on a MiSeq system (Illumina, San Diego, USA) with 300-bp paired-end reads. Image analysis and base calling were performed by sequence control software for real-time analysis and CASAVA software v1.8.2 (Illumina, San Diego, USA). These cleaned reads were aligned with the human reference genome sequence (UCSC hg19, NCBI build 37) using BWA (Version 0.7.12) with default parameters. The aligned read files in the BAM format were sorted and indexed using SAMtools. Somatic single-nucleotide variant and small indels calling was performed by the MuTect2 algorithm. Paired samples (cases 6 and 7) were analyzed by MuTect2 with the default setting, and the other tumor samples (cases 1 to 5) were analyzed in the tumor-only mode. Variants that passed the MuTect2 filters were annotated using ANNOVAR software [26]. Allele counting was performed using Integrative Genomics Viewer software (IGV).

## Sanger sequencing

The somatic variants detected by NGS were confirmed by Sanger sequencing. Fragments of *TP53*, *CTNNB1*, *ZNRF3*, and *PRKAR1A* were amplified by PCR using the primers previously reported (see S2 Table) [27,28,29]. Each 50 µl of PCR solution contained 25 µl of 2× PCR Buffer for KOD FX Neo, 10 µl of 2-mM dNTPs, 5 µl of 2-µM primer mix (0.2-µM final concentration of each primer), 1 µl of KOD FX Neo polymerase, 5 µl of DNA template (100–150 ng gDNA), and 5 µl of Nuclease-Free Water. We used the three-step touchdown protocol described above for PCR. The PCR products were purified with the QIAquick PCR Purification Kit (Qiagen, Hilden, Germany) and sequence reactions were performed using the BigDye Terminator Cycle Sequencing Kit v3.1 (Applied Biosystems, Foster, USA) according to the manufacturer's protocol. After purification using a DyeEx 2.0 Spin Kit (Qiagen, Hilden,

Germany), the products were separated on an Applied Biosystems 3130xl Genetic Analyzer (Applied Biosystems, Foster, USA) and the electropherograms were evaluated using Chromas Lite (http://www.technelysium.com.au/chromas_lite.html).

## Real-time quantitative PCR

Total RNA from seven cases of cortisol-producing ACC and seven cases of cortisol-producing ACA was obtained using the AllPrep DNA/RNA Mini Kit (Qiagen, Hilden, Germany). After extraction, the RNA concentration was determined by NanoDrop 1000 Spectrophotometer (Thermo Fisher, Waltham, USA) and an absorbance ratio greater than 2.0 for 260/280 nm was confirmed. One µg of RNA was used in each 20 µl reverse transcription reaction to cDNA. The reverse transcription reactions were performed with the SuperScript IV VILO Master Mix with ezDNase Enzyme (Thermo Fisher Scientific, Waltham, USA) according to the manufacturer's protocol. cDNA synthesized by reverse transcription reactions was diluted 1:2 with Nuclease-Free Water. For real-time quantitative PCR (RT-qPCR), TaqMan Gene Expression Master Mix (Applied Biosystems, Foster, USA) was used according to the manufacturer's protocol. Reactions used either 2 µL of diluted cDNA (50ng as template total RNA), as described above, in a 20-µL reaction and TaqMan assays (Thermo Fisher Scientific, Waltham, USA) (*CCNB2*: Hs00270424_m1, *AURKA*: Hs01582072_m1, and *GAPDH*: Hs99999905_m1) were used. The ΔΔCT method was used to evaluate the relative expression mRNA of targeted genes using GAPDH as an internal control.

## TCGA data collection

The ACC-TCGA (TCGA Provisional) dataset is publicly available through the Memorial Sloan-Kettering Cancer Center cBioPortal for Cancer Genomics (http://www.cbioportal). The dataset was obtained from the cBioPortal using a previously described method [30]. Of the 92 cases in the ACC-TCGA dataset from the cBioPortal, we analyzed the 79 cases in which mRNA sequencing was performed. Among them, 77 cases contained data on somatic variant, and 60 cases contained Weiss criteria evaluation results. We defined a Z-score > 0 as a high-expression group and ≤ 0 as a low-expression group in the mRNA expression data.

In addition, 90 out of 92 cases in the ACC-TCGA dataset were analyzed for somatic variants, of which 83 cases were also evaluated for hormone secretion. These 83 cases were evaluated for somatic variants of 12 candidate genes in this study and the presence or absence of cortisol production.

The term "mutation" shown in the ACC-TCGA dataset and in the figures created using Occo Print is equivalent to the "variants" described above in this article.

## Histological and immunohistochemical staining assessment

Immunohistochemistry analysis was performed using paraffin-embedded blocks of tissues obtained from patients during surgery. Pathologists at Tohoku University School of Medicine and/or Hamamatsu University School of Medicine hospital assessed Weiss criteria and Ki-67 index score. In this study, we defined specimens presenting three or more factors of Weiss histologic criteria as ACC.

## Statistical analysis

Associations between *TP53* or *CTNNB1* somatic variants and atypical mitotic figures were evaluated using the Fisher's exact test. For comparisons between the mRNA expression of two groups, the Mann–Whitney U test was used to determine differences. A significant difference

between mRNA expression levels for each pathological stage was performed using the Kruskal-Wallis test, and the post hoc analysis was performed using the Steel-Dwass test. Overall survival was calculated using the Kaplan-Meier method, and the significance of difference was analyzed using a log-rank test. The correlations between the mRNA expression of the two groups were analyzed by Spearman's rank correlation coefficient. All of the analyses mentioned above were performed with the EZR software [31], and $p < 0.05$ was considered to be statistically significant.

## Results

### Clinical characteristics

The clinical characteristics of the seven ACC cases are described in Table 1. The seven ACC cases included three men and four females. The median age at diagnosis of the patients was 47 years with a range of 20–57 years. The median of tumor size was 80 mm with a range of 55–152 mm. In all cases, cortisol production was not suppressed by a 1 mg or 8 mg DST. Therefore, the seven cases of ACC analyzed in this study were diagnosed as cortisol-producing ACC. Regarding hormones other than cortisol, case 2 showed estrogen, cases 3, 5, and 6 showed androgen, and case 7 showed aldosterone co-production. The staging of ACC patients was performed with the ENSAT 2008 staging system [9]. In this study, there were three cases of stage II, one case of stage III, and three cases of stage IV. The median follow-up period was 32 months with a range of 6–156 months and four of the seven cases were deceased.

### Somatic variant screening

The coverage summaries of deep sequencing for samples or target intervals were shown in S3 Table and S4 Table, respectively. The average coverage of the nine samples was 3501 X (ranged from 1570 X to 4827 X) (S3 Table) and most of the target regions were covered with more than X 200 coverages (S4 Table). However, we could not obtain enough read coverages in some target regions including exon 6 to 11 of the *MDM* gene in case 6 tumor and exon 9 to 11 of the *MDM* gene in case 7 blood.

 In case 6 and 7, paired data (from tumor and blood) were analyzed by MuTect2 to detect "*de novo*" somatic variants in tumor tissues. In other cases, we searched for possible pathogenic variants in tumor tissues using following exclusion criteria: 1) variants on mutant alleles with

**Table 1. Clinical characteristics of seven cases of cortisol-producing adrenocortical carcinoma.**

| Case | Sex | Age* | Tumor size(mm) | Hormone secreted | F (µg/dl) at DST | UFC (µg/day) | Stage | Outcome | Observation period (month) |
|------|-----|------|----------------|------------------|------------------|--------------|-------|---------|----------------------------|
| 1 | F | 50 | 55 | Cortisol | 19.3[a] | 1690 | 2 | Deceased | 32 |
| 2 | M | 54 | 143 | Cortisol, estrogen | 32.8[b] | 831 | 4 | Deceased | 37 |
| 3 | F | 25 | 150 | Cortisol, androgen | 40.3[b] | 1210 | 4 | Deceased | 13 |
| 4 | M | 46 | 80 | Cortisol | 17.2[a] | 79 | 2 | Alive | 156 |
| 5 | M | 47 | 152 | Cortisol, androgen | 14.0[b] | 187 | 4 | Deceased | 6 |
| 6 | F | 20 | 75 | Cortisol androgen | 20.3[a] | 271 | 3 | Alive | 34 |
| 7 | F | 57 | 61 | Cortisol, aldosterone | 15.3[a] | 251 | 2 | Alive | 6 |

* Age at diagnosis

[a] 1-mg DST

[b] 8-mg DST.

Abbreviations: F, female; M, Male; DST, dexamethasone suppression test; UFC, urine-free cortisol.

Staging performed according to ENSAT 2008.

less than 1% in the tumor tissue; 2) variants registered in the gnomAD and 3.5KJPN databases; 3) variants observed in our 218 in-house control exomes; 4) synonymous variants; and 5) variants predicted to likely be benign by multiple prediction tools. A total of seven somatic variants, two variants in *TP53* (c.375G>A:p.T125 = and c.749C>T:p.P250L), three variants in *CTNNB1* (c.110C>G:p.S37C, c.121A>G:p.T41A and c.133T>C:p.S45P), one variant in *PRKAR1A* (c.545C>A:p.T182K) and one variant in *ZNRF3* (c.433C>T:p.R145X), were identified in five of the seven ACC cases (71.4%, Fig 1). All variants identified by targeted deep sequencing using NGS were confirmed by Sanger Sequencing, as shown in S1 Fig. All variants were not registered in genome aggregation database (https://gnomad.broadinstitute.org/) and predicted to be deleterious by multiple *in silico* tools. The predicted pathogenicity of the candidate variants is summarized in S5 Table. Somatic variants in *TP53*, *CTNNB1*, and *ZNRF3* had been reported as pathogenic variants in cases of ACC or other cancers [32, 33, 34]. Although the somatic variant in *PRKAR1A* (NM_002734.4):c.545C>A (p.T182K) is a novel missense variant. This variant was predicted to be deleterious by multiple algorithms for pathogenicity prediction: the scale-invariant feature transform score was 0.005, combined annotation-dependent depletion PHRED score was 25.3, Phastcons score was 1.0, and phylphen2 score was 0.009. Sanger sequencing using paired tumor and blood DNA confirmed that this variant occurred *de novo* in the tumor (S1 Fig).

## Combined analysis of somatic variants and pathological findings

The Weiss score, positive or negative result of each Weiss criteria, and Ki-67 index are shown in Fig 1. Furthermore, we performed a combined analysis of somatic variants and those pathological findings. All of the cases with a Ki-67 index ≥ 10% had *TP53* or *CTNNB1* somatic

| Case | Gene | Somatic variant | Mut/Ref counts[a] (%[b]) | | Ki-67 Index (%) | Weis score | Weiss criteria | | | | | | | | |
|---|---|---|---|---|---|---|---|---|---|---|---|---|---|---|---|
| | | | Tumor Tissue | Blood | | | Nuclear atypia | Mitotic rate | Atypical mitotic figures | Clear cells ≤ 25% | Diffuse architecture | Necrosis | Venous invasion | Sinusoidal invasion | Capsular invasion |
| 1 | | not detected | | N.A. | 5.5 | 6 | 1 | 0 | 0 | 1 | 1 | 0 | 1 | 1 | 1 |
| 2 | *TP53* *ZNRF3* | c.375G>A:p.T125= c.433C>T:p.R145X | 962/98 (90.7) 466/136 (77.4) | N.A. | 60.0 | 8 | 1 | 1 | 1 | 1 | 1 | 1 | 1 | 0 | 1 |
| 3 | *CTNNB1* *TP53* | c.110C>G:p.S37C c.749C>T:p.P250L | 1,482/4,281 (25.7) 2,357/480 (83.1) | N.A. | 15.8 | 7 | 1 | 1 | 1 | 1 | 1 | 1 | 0 | 0 | 1 |
| 4 | | not detected | | N.A. | 4.0 | 3 | 0 | 0 | 0 | 0 | 1 | 1 | 0 | 1 | 0 |
| 5 | *CTNNB1* | c.121A>G:p.T41A | 2,561/2,762 (48.1) | N.A. | 14.8 | 8 | 1 | 1 | 0 | 1 | 1 | 1 | 1 | 1 | 1 |
| 6 | *PRKAR1A* | c.545C>A:p.T182K | 1,985/408 (82.9) | 0/2,455 (0) | 6.0 | 6 | 1 | 1 | 0 | 1 | 1 | 0 | 0 | 1 | 1 |
| 7 | *CTNNB1* | c.133T>C:p.S45P | 1,927/2,929 (39.7) | 0/6,952 (0) | 13.0 | 6 | 1 | 1 | 0 | 1 | 1 | 1 | 0 | 0 | 1 |

**Fig 1. Clinical characteristics of seven cases of cortisol-producing adrenocortical carcinoma.** Shows a detailed comparison of the somatic variant identified by NGS with Ki-67 index (%) and Weiss criteria. *In the Weiss criteria columns, 0 and 1 signify negative and positive findings for each factor, respectively. Mut, Mutant allele; Ref, Reference allele; N.A., not assessed; [a]; Variants were called by MuTect2, and reads were manually counted by IGV. [b]; Percent of mutant allele was calculated by allele reads/(mutant allele reads + reference allele reads).

variants, both previously reported as poor prognosis factors. In contrast, cases with Ki-67 index < 10% had no *TP53* or *CTNNB1* somatic variants. We compared findings for each Weiss criterion in cases with *TP53* or *CTNNB1* somatic variants and Ki-67 index ≥ 10% and the others. In our cases, atypical mitotic figures were positive in all of the cases with *TP53* somatic variants and Ki-67 index ≥ 10%.

In contrast, atypical mitotic figures were negative in cases without *TP53* somatic variants. From these results, we hypothesized that in cases with *TP53* somatic variants, an abnormality occurred in the M phase of the cell cycle, causing an atypical mitotic figure to appear. To verify the hypothesis, we decided to use the ACC-TCGA dataset obtained from cBioPortal because our study contained very few cases for analysis.

## Verification of association with *TP53* somatic variant and atypical mitotic figures from ACC-TCGA dataset

First, we confirmed variants of 12 candidate genes in cortisol-producing ACC in the ACC-TACG dataset. This dataset contained 37 cases of cortisol-producing ACC (20 cases of androgen and cortisol, 16 cases of cortisol, and 1 case of aldosterone and cortisol) among the 83 cases that evaluated hormone secretion. In those 37 cortisol-producing ACC cases, we investigated the somatic variants of the 12 candidate genes that we targeted for NGS. We identified nine *CTNNB1* variant cases, seven *TP53* variant cases, five *MEN1* variant cases, three *PRKAR1A* variant cases, two *ZNRF3* variant cases, one *APC* variant case, one *CDK2A* variant case, one *MDM2* variant case, and one *RB1* variant case (S2 Fig). Out of 37 cases, 21 (56.8%) harbored somatic variants in at least 1 of the 12 candidate genes. Thirteen cases (35.1%) harboring either *CTNNB1*, *ZNRF3*, or *PRKAR1A* somatic variants involved cortisol production. Conversely, 1 of the three somatic variants were found in 10 (21.7%) of 46 non-cortisol-producing ACCs.

For additional examination of the association between *TP53* somatic variant and atypical mitotic figures, we selected 79 ACC-TCGA dataset cases where RNA sequencing was performed. Of the 79 analyzed cases, 16 cases (20.3%) harbored *TP53* somatic variants and 13 cases (16.5%) harbored *CTNNB1* somatic variants. Of 60 cases with complete Weiss criteria, 31 cases (51.2%) presented atypical mitotic figures. A summary of the 79 cases analyzed in this study is presented in Fig 2. Next, associations between *TP53* or *CTNNB1* somatic variants and atypical mitotic figures were evaluated using Fisher's exact test. As shown in Table 2A, among 14 cases with *TP53* somatic variant, 12 (85.7%) showed atypical mitotic figures. Significantly more atypical mitotic figures were present in cases with *TP53* somatic variant (p < 0.001). By contrast, associations between *CTNNB1* somatic variant and atypical mitotic figures (Table 2B) were not significantly different (p = 1.00). Five cases had *CTNNB1* somatic variant and atypical mitotic figures, of which three cases also harbored *TP53* somatic variants and one case had observed copy-number alterations of the gene belonging to the TP53-RB1 pathway. In the *TP53* somatic variant cases, the results were consistent with the hypothesis that atypical mitotic figures appear through M phase abnormalities. Though both *TP53* and *CTNNB1* variants have been reported as poor prognostic factors, we speculated that the two somatic variants produced different abnormal cell cycle phases.

## Analysis of mRNA expression involved in M phase of cell cycle in ACC-TCGA dataset

It has been reported that *CKD1*, *CCNB1* (*cyclin B1*), *CCNB2* (*cyclin B2*), *CDC25C*, and *TOP2A* are more frequently expressed in ACC than in ACA as factors involved in the G2/M phase [35]. In addition, *AURAKA* (Aurora kinase A) has been reported to cause atypical mitosis due to overexpression [36]. Of these, we evaluated whether *CDK1*, *CCNB1*, *CCNB2*, and *AURKA*

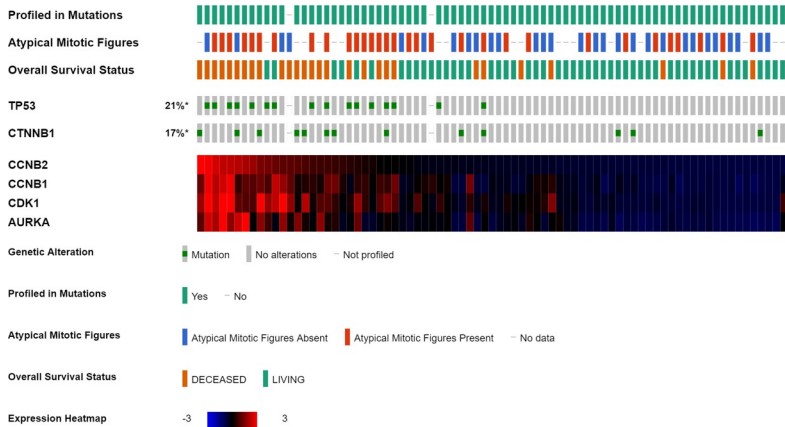

**Fig 2. Summary of ACC-TCGA dataset used in this study.** Among the cases in the adrenocortical carcinoma dataset (TCGA Provisional), these are the 79 cases for which summaries were analyzed in this study. The presence or absence of *TP53* or *CTNNB1* variants; the presence or absence of atypical mitotic figures; and heatmaps of mRNA expression (Z-score) of *CCNB2*, *CCNB1*, *CDK1*, and *AURKA* were summarized using Occo Print. "Mutation" in Occo Print is synonymous with "variant" in this article.

mRNA expression, which has a particular activity during the M phase or atypical mitosis, were different depending on the presence or absence of *TP53* or *CTNNB1* somatic variants. Analysis of the ACC-TCGA dataset revealed that the *TP53* mutated group had significantly higher mRNA expression in *CDK1* ($p < 0.001$), *CCNB1* ($p < 0.001$), *CCNB2* ($p < 0.001$), and *AURKA* ($p < 0.001$) compared with the *TP53* wild group (Fig 3A, 3B, 3C and 3D). Compared with the wild group, the *CTNNB1* mutated group showed significantly higher mRNA expression levels only in the *CCNB2* ($p = 0.046$) group, but no significant difference in the *CDK1* ($p = 0.116$), *CCNB1* ($p = 0.194$), and *AURAKA* ($p = 0.126$) groups (Fig 3E, 3F, 3G and 3H).

In the box plots, bounds of the box span from the first quartile (Q1) to the third quartile (Q3), and the center line represents the median. The lower whisker extends up to [Q1 − 1.5 × (Q3 − Q1)] and upper whisker extends up to [Q3 + 1.5 × (Q3 − Q1)]. Statistical analysis: $^{**}p < 0.01$, $^{*}p < 0.05$ (Mann–Whitney U test).

Furthermore, a significant difference in the mRNA expression level of *CCNB2* was absent in the *CTNNB1* mutated group when the four cases with simultaneous *TP53* variants were excluded ($p = 0.191$) (S3 Fig). *TP53* mutated cases are likely to have abnormalities in the M phase, but *CTNNB1* mutated cases are likely not to have abnormalities in the M phase.

**Table 2. A**. Associations between *TP53* variants and atypical mitotic figures. **B**. Associations between *CTNNB1* variants and atypical mitotic figures.

**A**

| Group | Atypical mitotic figures absent (%) | Atypical mitotic figures present (%) | Total (%) |
|---|---|---|---|
| *TP53* mutated | 2 (3.4) | 12 (20.7) | 14 (24.1) |
| *TP53* wild | 26 (44.9) | 18 (31.0) | 44 (75.9) |
| total | 28 (48.3) | 30 (51.7) | 58 (100) |

**B**

| | | | |
|---|---|---|---|
| *CTNNB1* mutated | 4 (6.9) | 5 (8.6) | 9 (15.5) |
| *CTNNB1* wild | 24 (41.4) | 25 (43.1) | 49 (84.5) |
| total | 28 (48.3) | 30 (51.7) | 58 (100) |

$p < 0.001$ (Fisher's exact test). $p = 1.00$ (Fisher's exact test).

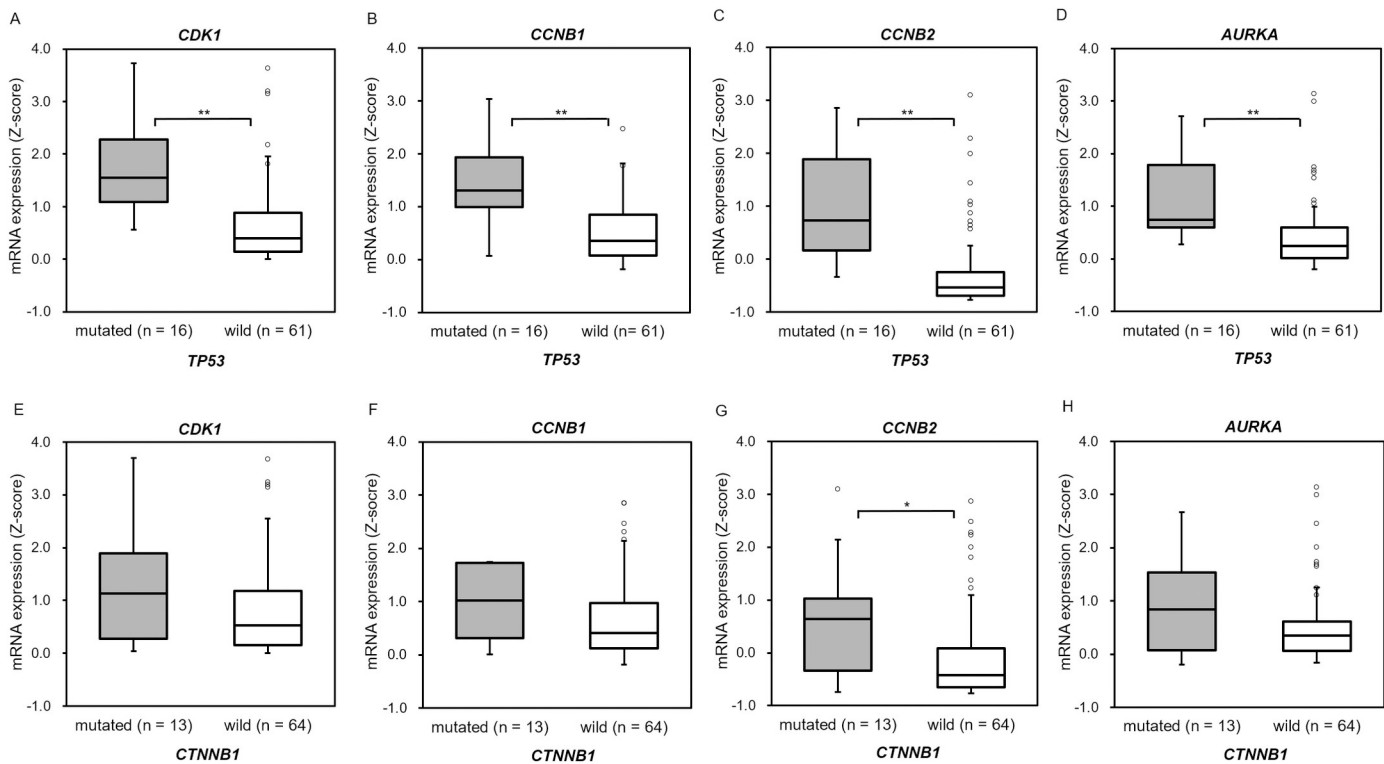

**Fig 3. *TP53* mutated cases in the ACC-TCGA dataset showed increased mRNA expression involved in the M phase.** (A-D) Comparison of *CDK1* (A), *CCNB1* (B), *CCNB2* (C), and *AURKA* (D) mRNA expression in *TP53* mutated cases and *TP53* wild type cases. (E-H) Comparison of *CDK1* (E), *CCNB1* (F), *CCNB2* (G), and *AURKA* (H) mRNA expression in *CTNNB1* mutated cases and *CTNNB1* wild type cases. *TP53* mutated cases had significantly higher mRNA expression in *CDK1* (p < 0.001), *CCNB1* (p < 0.001), *CCNB2* (p < 0.001), and *AURKA* (p < 0.001) groups. *CTNNB1* mutated cases showed significantly higher mRNA expression levels only in the *CCNB2* (p = 0.046) group.

We then examined whether there is a difference in the mRNA expression of *CDK1*, *CCNB1*, *CCNB2*, and *AURKA* depending on the presence or absence of atypical mitotic figures. In cases with atypical mitotic figures, mRNA expression of *CCNB1* (p = 0.068) was not significantly different from that of cases without atypical mitotic figures, although the mRNA of *CDK1* (p = 0.029), *CCNB2* (p = 0.012), and *AURKA* (p = 0.018) was significantly highly expressed (Fig 4A, 4B, 4C and 4D). These results indicated that *TP53* somatic variant and atypical mitotic figures are associated with high mRNA expression of *CDK1*, *CCNB2*, and *AURKA*.

In the box plots, bounds of the box span from the first quartile (Q1) to the third quartile (Q3), and the center line represents the median. The lower whisker extends up to [Q1 − 1.5 × (Q3 − Q1)] and upper whisker extends up to [Q3 + 1.5 × (Q3 − Q1)]. Statistical analysis: *p < 0.05 (Mann–Whitney U test).

We further investigated whether overexpression of *CDK1*, *CCNB2*, or *AURKA* mRNA was a factor in poor prognosis for ACC. We compared the mRNA expression levels of *CDK1*, *CCNB2*, and *AURKA* at different ACC pathological stages. The expression of *CDK1* and *AURKA* was significantly higher at stage IV than at stages I (p = 0.003 and p = 0.024) and II (p < 0.001 and p = 0.024). The expression of *CCNB2* was significantly higher at stage IV than at stages I (p = 0.034) and II (p < 0.001) and higher at stage III than at stage II (p = 0.027) (Fig 5)

Next, we used the ACC-TCGA dataset to define mRNA expression of *CDK1*, *CCNB2*, and *AURKA*, where a Z-score > 0 was the high-expression group and Z-score ≤ 0 was the low-expression group. Then, the overall survival between the two groups was compared (Fig 6).

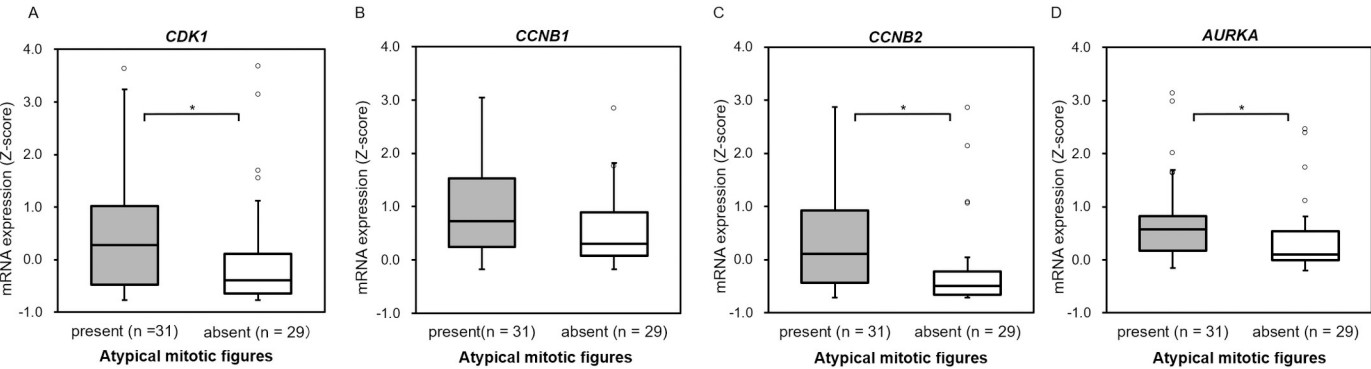

**Fig 4. Cases with atypical mitotic figures in the ACC-TCGA dataset showed increased mRNA expression of *CDK1*, *CCNB2*, and *AURKA*.** (A-D) Comparison of *CDK1* (A), *CCNB1* (B), *CCNB2* (C), and *AURKA* (D) mRNA expression in cases with present with atypical mitotic figures and cases with absent with atypical mitotic figures. Cases with atypical mitotic figures showed that mRNA expression of CDK1 (p = 0.029), CCNB2 (p = 0.012), and AURKA (p = 0.018) was significantly more highly expressed.

Overall survival was significantly shorter in the high-expression group than in the low-expression group in all *CDK1*, *CCNB2*, and *AURKA* groups (p < 0.001).

In the box plots, bounds of the box span from the first quartile (Q1) to the third quartile (Q3), and the center line represents the median. The lower whisker extends up to [Q1 − 1.5 × (Q3 − Q1)] and upper whisker extends up to [Q3 + 1.5 × (Q3 − Q1)].

These results indicated that M phase abnormalities due to overexpression of *CDK1*, *CCNB2*, and *AURKA* are factors indicating a poor prognosis for ACC.

In addition to the above results, we examined the correlation between the mRNA expression of *MIKI67* (which encodes the Ki-67 protein) and *CDK1*, *CCNB2*, and *AURKA* because the ACC-TCGA dataset contains no Ki-67 index. The mRNA expression of *MIKI67* and *CDK1* (Spearman's rank $r_s$ = 0.95, p < 0.001), *CCNB2* ($r_s$ = 0.85, p < 0.001), and *AURAKA* ($r_s$ = 0.84, p < 0.001) were all positively correlated (S4 Fig). This result suggests that the mRNA expression levels of *CDK1*, *CCNB2*, and *AURKA* may be a prognostic predictor as well as the Ki-67 index, which is the most commonly used prognostic predictor for ACC.

## Evaluation of mRNA expression of *CCNB2* and *AURKA* in seven cortisol-producing ACC cases

Analysis of mRNA expression by RT-qPCR was applied to the seven cases of cortisol-producing ACC subjected to genetic analysis this study, as well as seven cases of cortisol-producing ACA

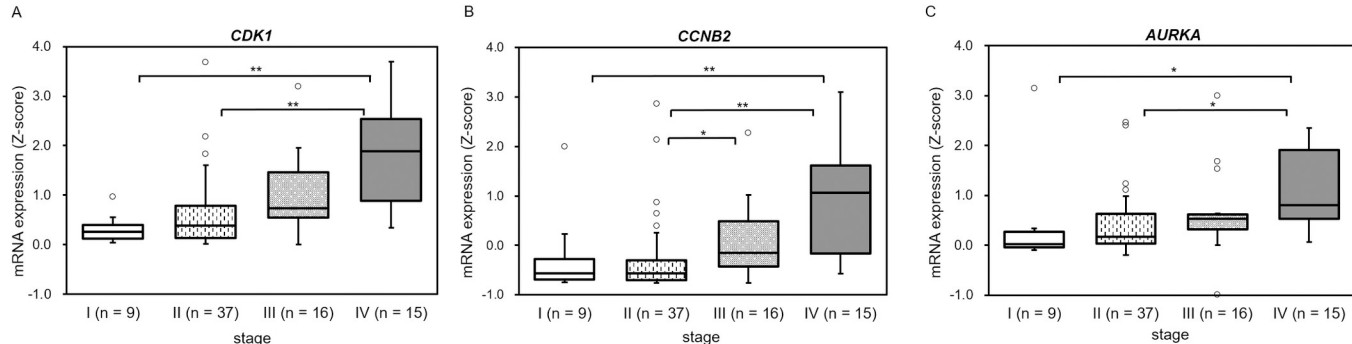

**Fig 5. Significant correlations between *CDK1*, *CCNB2*, and *AURKA* mRNA expression with pathological stage.** (A-C) Significant correlations between expression levels of *CDK1* (A), *CCNB2* (B), and *AURKA* (C) for different ACC pathological stages. Significant differences were observed in all groups of *CDK1*, *CCNB2*, and *AURKA* comparisons using the Kruskal-Wallis test (P < 0.01). The Steel-Dwass test was used for post hoc analysis (**p < 0.01, *p < 0.05).

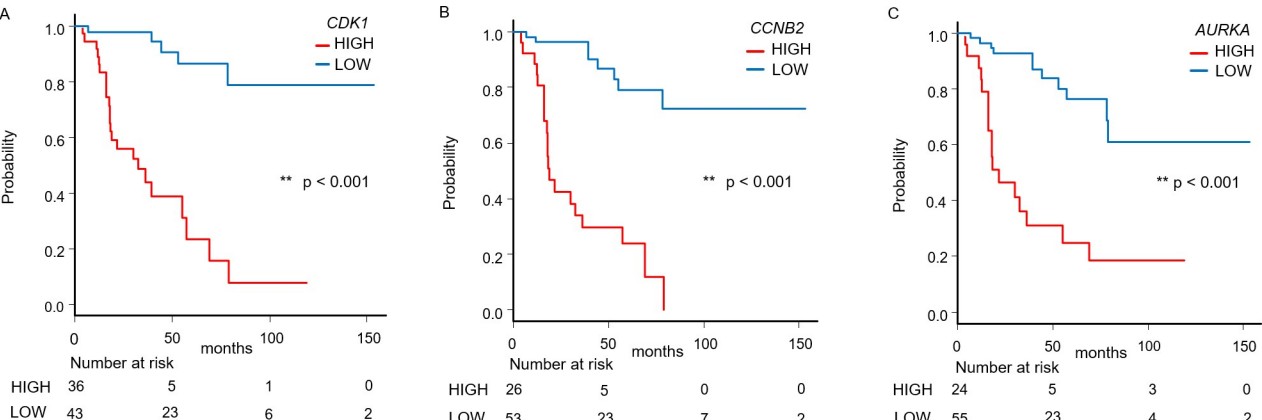

**Fig 6. Significant correlations between *CDK1*, *CCNB2*, and *AURKA* mRNA expression with survival.** (A-C) Survival analysis of *CDK1 (A)*, *CCNB2 (B)*, and *AURKA* (C) in ACC. This study analyzed 79 cases from the TCGA-ACC dataset. The mRNA expression Z-score (RNA Seq V2 RSEM) > 0 was defined as the high-expression group and Z-score ≤ 0 as the low-expression group. The difference between overall survival in the high-expression group and the low-expression group was compared for *CDK1*, *CCNB2*, and *AURKA* using the Kaplan-Meier method. Overall survival was significantly shortened in all high-expression *CDK1*, *CCNB2*, and *AURKA* groups. The p-value was calculated using the log-rank test.

used for comparison. We examined *CCNB2* and *AURKA*, which were strongly related to *TP53* and atypical mitotic figures in the ACC-TCGA dataset analysis. The relative mRNA expression level of *CCNB2* was 6.56-fold higher in ACC than in ACA (p = 0.018) (Fig 7A). In particular, in cases 2 and 3 with *TP53* somatic variants, the relative mRNA expression of *CCNB2* was 31.3- and 21.7-fold higher (Fig 7B), respectively. The relative mRNA expression level of *AURKA* was 3.38-fold higher in ACC than in ACA, but it was not statistically significantly different (p = 0.097) (Fig 7C). However, the relative mRNA expression level of *AURKA* was higher in cases showing atypical mitotic figures (16.2-fold in case 2, 8.3-fold in case 3) (Fig 7D).

## Discussion

In our study of Japanese cases of cortisol-producing ACC, 5 of 7 cases (71.4%) showed somatic variants in at least 1 of 12 candidate genes. The five cases in which somatic variants were identified in this study harbored somatic variants in either *CTNNB1*, *ZNRF3*, or *PRKAR1A*, which are involved in cortisol production. These somatic variants related to cortisol production may be involved in the poor prognosis of cortisol-producing ACC. Although the *CTNNB1* variant has been studied [12,16], further verification is required for the *ZNRF3* and *PRAKR1A* variants. Although our study is not definitive because the number of cases analyzed was small, the frequency of detection of somatic variants involved in cortisol production in our cases was higher than that of cortisol-producing ACC cases in the ACC-TCGA dataset. One possible reason for the high frequency of somatic variants involved in cortisol production in our study is that the cortisol level of the cases we analyzed was higher, although it was difficult to compare because no cortisol levels are reported in the ACC-TCGA dataset.

In this study, a novel variant, *PRKAR1A* (NM_002734.4);c.545C>A (p.T182K), was detected. *PRKAR1A* has been reported as a driver gene of ACC in a previous report [12], and germline variant is known as the cause for the Carney complex, or primary pigmented nodular adrenocortical disease (PPNAD) [37,38]. In *PRKAR1A* variants, 80% of variants, such as frameshift or nonsense variants (with premature stop codon), are subject to mRNA nonsense-mediated decay (NMD), which prevents translation of proteins. However, 20% of the variants that escaped NMD and were translated into a protein have been reported to cause more

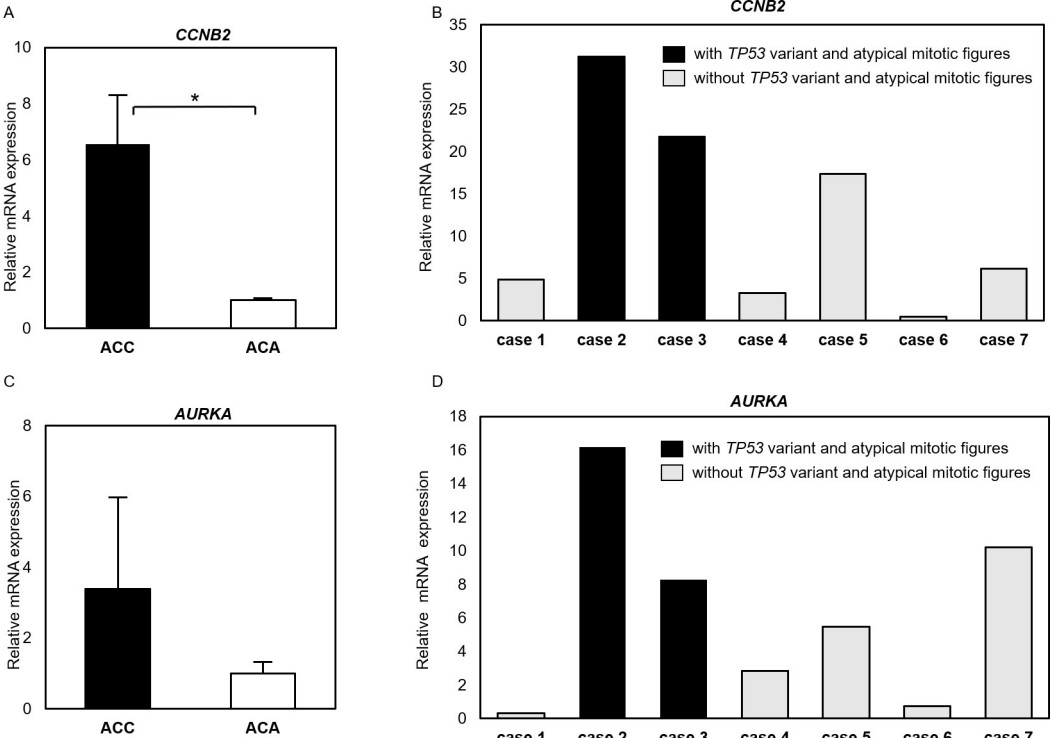

**Fig 7. Analysis of mRNA expression by RT-qPCR of *CCNB2* and *AURKA* in the cases examined in this study.** (A) Expression of *CCNB2* mRNA in ACC was significantly higher than in ACA (p = 0.018). (B) mRNA expression of *CCNB2* in each case of ACC. In cases 2 and 3 with *TP53* somatic variants and atypical mitotic figures, *CCNB2* expression was 31.3- and 21.7-fold higher than in ACA, respectively. (C) The expression level of *AURKA* mRNA had no statistically significant difference between the ACC and ACA groups (p > 0.05). (D) mRNA expression of *AURKA* in each ACC case. In cases 2 and 3 with *TP53* somatic variants and atypical mitotic figures, *AURKA* expression was 16.2- and 8.3-fold higher than in ACA, The relative mRNA expression level of *CCNB2* or *AURKA* in ACC was calculated based on 2-ΔΔCt, expressed as a multiple of the ACA value as a control and using *GAPDH* as a reference gene. Error bars express the standard error of the mean. Statistical analysis: *p < 0.05 (Mann–Whitney U test).

aggressive tumors due to the dominant-negative effect [37,39]. The c.545C>A (p.T182K) somatic variant located at the cAMP binding domain A of PRKAR1A protein is likely to escape NDM. Anselmo et al. reported that a germline c.439A>G (p.S147G) variant in the cAMP binding domain A caused ACC in a woman [40]. In addition, three germline *PRKAR1A* variants in the cAMP binding domain A (c.547G>T (p.A183Y), c.638C>A (p.213D), and c.438A>T (p.R146S)) had been reported and the functional studies *in vitro* have demonstrated that these missense variants might enhance the PKA activity leading to accelerating the tumor genesis. It has been proposed that this PKA activity enhancement is caused by an increase in cAMP-specific PKA activity through conformational changes in the cAMP binding domain, despite a decrease in cAMP binding of PRKAR1A [41,42]. Although the *PRKAR1A*:c.545C>A (p.T182K) variant is considered as "variant of uncertain significance" at this time, we hypothesized that the c.545C>A (p.T182K) variant was also likely to cause the upregulation of PKA activity and involve in ACC carcinogenesis. However, details of the involvement of this *PRKAR1A* somatic variant in carcinogenesis of the adrenal gland require further investigation.

*ZNRF3* is the most frequently altered gene in ACC, and it is well known that copy number changes are frequent [14]. *ZNRF3* has been shown to act as a tumor suppressor, promoting Wnt receptor turnover. Inhibition of *ZNRF3* enhances Wnt pathway signaling and is involved in tumorigenesis [43]. However, *ZNRF3* somatic variants were less frequent than copy number

changes and were reported in 7 of 77 ACC cases in previous reports [15]. Four of the seven cases are nonsense variants that affect sequences upstream of the transmembrane domain, and it has been speculated those nonsense variants lead producing to a truncated protein unable to anchor properly to the cell membrane. The *ZNRF3*:c.433C>T (p.R145X) variant identified in this study is also a nonsense variant located at a site presumed to affect the sequence upstream of the transmembrane domain. The *ZNRF3*:c.433C>T (p.R145X) variant is speculated to be involved in carcinogenesis by upregulating the Wnt pathways.

Among the Weiss criteria factors, the mitotic rate has been reported to be associated with patient outcome [5]. In an analysis of our cases, it was speculated that cases with *TP53* somatic variants were likely to show a Ki-67 index > 10% and atypical mitotic figures. Morimoto et al. examined Ki-67 index findings and Weiss criteria findings in 17 cases of ACC [7]. In the study, 50% (two out of four) of cases with a Ki-67 index > 10% presented atypical mitosis figures. Conversely, in cases with a Ki-67 index < 10%, atypical mitosis figures were absent. These results indicate that ACC cases with atypical mitotic figures may be more aggressive. When we verified our hypothesis using the ACC-TCGA dataset, the frequency of atypical mitotic figures increased significantly in cases with *TP53* somatic variant, but no significant increase was observed in cases with the *CTNNB1* somatic variant.

The analysis using the ACC-TCGA dataset showed that the appearance of atypical mitotic figures in *CTNNB1* mutated cases was likely to be affected by TP53-RB1 pathway alterations. In addition, verification of mRNA expression by the ACC-TCGA dataset suggested that abnormalities in genes related to the M phase were not involved in the poor prognosis of *CTNNB1* mutated cases. While *CTNNB1* mutated ACC is considered to be a poor prognostic factor, it has been reported that cortisol-producing ACA also has *CTNNB1* variants with the same frequency as that for ACC [44]. In this study, the mechanism of the poor prognosis of *CTNNB1* mutated cases could not be clarified. However, if the mechanism of poor prognosis in *CTNNB1* mutated ACC cases is elucidated in the future, the pathogenesis of ACC will be clarified, and the development of personalized ACC medicine can be expected to advance.

In contrast, in the cases with *TP53* somatic variant in which the frequency of atypical mitotic figures significantly increases, it was found that the mRNA of *CDK1*, *CCNB1*, *CCNB2*, and *AURKA*, which are involved in the regulation of the M phase, was significantly highly expressed. Furthermore, the mRNA of *CDK1*, *CCNB2*, and *AURKA* was significantly highly expressed in the cases with atypical mitotic figures. Because CDK1 is active only after binding to cyclin B, high expression of *CCNB2* and *AURKA* can be considered to be involved in atypical mitosis.

The mechanism of atypical mitosis is still unclear, but there are reports that centrosome abnormalities are involved [45], and the centrosome is associated with G2/M checkpoint regulation [46].

Overexpression of *CCNB2* has been revealed in cancers such as breast cancer and lung cancer, and it has been reported as a poor prognostic factor [47,48]. In ACC, it is reported that *CCNB2* is 5.6- to 14-fold more highly expressed compared with ACA by microarray analysis and that *CCNB2* is overexpressed in more aggressive forms of ACC [35,49,50,51]. RT-qPCR analysis in our cases also showed a high expression of *CCNB2* mRNA in ACC compared with expression in ACA. In particular, expression was higher in *TP53* somatic variant cases. Our study showed that high *CCNB2* expression in ACC was associated with *TP53* somatic variants and atypical mitotic figures. Nam et al. reported that CCNB2 and P53 act antagonistically to control AURKA-mediated centrosome splitting and accurate chromosome segregation in normal cells [52]. This antagonistic AURKA-mediated relationship between CCNB2 and P53 may explain why *TP53* somatic variants were associated with increased atypical mitotic figures in our study and may also point to a new treatment strategy for some ACC cases. Borges et al.

reported that overexpression of *AURKA* occurs in pediatric ACC with the *TP53* p.R337H variant [53], and that AMG900, an Aurora kinase inhibitor, acted synergistically with mitotane and doxorubicin in the inhibition of H295R cell proliferation [54]. The RT-qPCR results of our seven cases of cortisol-producing ACC and the results of ACC-TCGA dataset analysis showed that mRNA of *AURKA* is also highly expressed in adult ACC cases with *TP53* somatic variants and with atypical mitotic figures. Alisertib, an AURKA inhibitor, is currently undergoing clinical trials for various cancers [55,56,57], and it may be useful for the treatment of ACC in the future. In aggressive cases with *TP53* variants, it may be more effective to administer an AURKA inhibitor in combination with conventional mitotane and combined etoposide, doxorubicin, and cisplatin therapy as adjuvant therapy. Mo et al. reported that the relative expression level of circulating *CCNB2* mRNA in cancer patients (not including ACC) was significantly higher than that in normal controls and a group with benign diseases, and that expression levels of circulating *CCNB2* mRNA in cancer patients significantly decreased after treatment [58]. In adult ACC, *TP53* variants do not have an identified hotspot and are not concentrated in specific exons, so it is currently challenging to confirm the presence or absence of *TP53* variants for all cases in clinical practice. In the future, if it is possible to measure the serum circulating mRNA of *CCNB2* easily, this may be useful for selecting cases to be screened for *TP53* variants in ACC, determining therapeutic effects, and predicting recurrence. Because biopsy is contraindicated in ACC, it is challenging to identify somatic variants of *TP53* and confirm atypical mitotic figures in inoperable cases. Measurement of expression levels of circulating *CCNB2* mRNA may be helpful for selecting AURKA inhibitor treatment for inoperable cases of ACC.

The presence or absence of atypical mitotic figures is a factor that is unlikely to be a problem of inter-individual reproducibility in the Weiss criteria [59], and it is a factor that can also be used in current clinical practice. In this study, we showed that ACC with atypical mitosis harbored *TP53* somatic variants and is likely to be associated with AURKA-mediated M phase deregulation due to overexpression of *CCNB2*. When AURKA inhibitors can be used clinically in the future, ACC with atypical mitotic figures may be an active indication for AURKA inhibitor administration.

## Study limitations

In this study, the limitations were that the number of cases was small and that blood samples could be used as a reference in only two cases. In cancer, low-frequency allelic variant may affect carcinogenicity. The lack of reference blood may mean that the advantages of targeted deep sequencing were not fully exploited to identify low-frequency allelic variants. Furthermore, because this study mainly focused on somatic variants of the TP53-RB1 and Wnt pathways, crosstalk of alterations with other pathways has not been fully verified. Additional validation was performed using the ACC-TCGA dataset to supplement the low number of cases. Despite this, the problem of the small number of cases compared with other carcinomas has not been solved, and the ACC-TCGA dataset did not have Ki-67 index data. In addition, a major limitation of this study is that protein functions, including phosphorylation, were not evaluated.

## Conclusion

We conducted targeted deep sequencing using NGS for seven Japanese patients with cortisol-producing ACC. *PRKAR1A* is involved in cortisol production and has been reported as a driver gene for ACC. We identified a novel somatic variant of *PRKAR1A* c.545C>A (p. T182K). The effect of this variant on carcinogenesis needs further investigation. In addition, we reported, for what we believe to be the first time, that ACC with *TP53* somatic variants is frequently associated with atypical mitotic figures and a higher expression of *CCNB2* and

*AURKA*. In the future, when AURKA inhibitors such as Alisertib can be used clinically, it may be useful to use the presence or absence of atypical mitotic figures as an index for drug administration in daily clinical practice.

## Supporting information

**S1 Fig. Confirmation of somatic variants identified by NGS using Sanger sequencing.** Sanger sequencing results are shown under the reference nucleotide sequences. The upper electropherograms show the sequencing results from the ACC sample, and arrows indicate the altered nucleotides. The lower electropherograms show the sequencing results for the references. Cases 6 and 7 used the patients' own blood samples as references (indicated as blood DNA in the figure), and in other cases, healthy adult blood samples were used as the references (indicated as control DNA in the figure). (A) *ZNRF3* c.433C>T (p.R145X) variant in case 2, (B) *TP53* c.375G>A (p.T125 = ) variant in case 2, (C) *CTNNB1* c.110C>G (p.S37C) variant in case 3, (D) *TP53* c.749C>T (p. P250L) variant in case 3, (E) *CTNNB1* c.121A>G (p.T41A) variant in case 5, (F) *PRKAR1A* c.545C>A (p.T182K) variant in case 6, and (G) *CTNNB1* c.133T>C (p.S45P) variant in case 7.
(TIF)

**S2 Fig. ACC-TCGA dataset summary for cortisol-producing ACC.** Occo Print was used to evaluate hormone excess in 83 cases of ACC from the TCGA Provisional dataset and summarize the presence or absence of variants in the 12 candidate genes of this study. "Mutation" in Occo Print is synonymous with "variant" in this article.
(TIF)

**S3 Fig. Comparison of expression of *CCNB2* mRNA in *CTNNB1* mutated cases (excluding *TP53* co-mutated 4 cases) and wild type.** When the analysis was performed excluding 4 cases of *TP53* co-mutated cases, the significant difference in *CCNB2* mRNA expression between *CTNNB1* mutated cases and wild type disappeared (P = 0.191). In the box plots, bounds of the box span from the first quartile (Q1) to the third quartile (Q3), and the center line represents the median. The lower whisker extends up to $[Q1 - 1.5 \times (Q3 - Q1)]$ and upper whisker extends up to $[Q3 + 1.5 \times (Q3 - Q1)]$.
(TIF)

**S4 Fig. *MKI67* mRNA expression associated with mRNA expression involved in the M phase of ACC** (A) *CDK1* mRNA expression in ACC was positively correlated with that of *MIKI67* mRNA expression (Spearman's rank, $r_s = 0.95$, $p < 0.001$). (B) *CCNB2* mRNA expression in ACC was positively correlated with that of *MIKI67* mRNA expression ($r_s = 0.85$, $p < 0.001$). (C) *AURKA* mRNA expression in ACC was positively correlated with that of *MIKI67* mRNA expression ($r_s = 0.84$, $p < 0.001$).
(TIF)

**S1 Table. Primer list of 12 candidate genes for targeted deep sequencing** Forward and reverse (5' to 3') primers and product size and PCR conditions are shown in the table.
(XLSX)

**S2 Table. Primer list for Sanger sequencing.** These primers were used to confirm the variants detected by NGS. PCR was performed using the three-step touchdown PCR protocol.
(XLSX)

**S3 Table. Summary of NGS coverage information for each sample.** Cases 1–7 (tumor) are tumor samples of ACC, and cases 6 and 7 (blood) are reference blood samples for cases 6 and 7 (tumor). The notation > 100 (%), 200 (%), and 500 (%) indicates the percentage by which

the number of coverages exceed 100, 200, and 500, respectively, in each sample.
(XLSX)

**S4 Table. Summary of NGS coverage information for each target regions.** The
notation > 100 (%), 200 (%), and 500 (%) indicates the percentage by which the number of
coverages exceeds 100, 200, and 500, respectively, in each target regions.
(XLSX)

**S5 Table. Summary of predicted pathogenicity of candiate variants.** The SIFT, PolyPhen2
HumVar, CADD phred, GERP, and PhastCons scores for each variant detected in this study
are shown. It shows whether these variants are registered in gnomAD, COSMIC, TCGA, and
ClinVar. It also describes the tier classification of each variant in Cancer Gene Census and the
oncogenic in OncoKB of each variant.
(XLSX)

## Author Contributions

**Conceptualization:** Akira Ikeya, Miho Yamashita.

**Data curation:** Akira Ikeya, Keisuke Kakizawa, Yuta Okawa.

**Formal analysis:** Mitsuko Nakashima, Hironobu Sasano.

**Funding acquisition:** Mitsuko Nakashima.

**Investigation:** Akira Ikeya, Mitsuko Nakashima.

**Methodology:** Akira Ikeya, Mitsuko Nakashima, Miho Yamashita.

**Project administration:** Miho Yamashita, Yutaka Oki.

**Resources:** Miho Yamashita.

**Supervision:** Hirotomo Saitsu, Shigekazu Sasaki, Hironobu Sasano, Takafumi Suda, Yutaka
Oki.

**Validation:** Akira Ikeya, Mitsuko Nakashima, Keisuke Kakizawa, Yuta Okawa.

**Visualization:** Akira Ikeya, Mitsuko Nakashima.

**Writing – original draft:** Akira Ikeya, Mitsuko Nakashima.

**Writing – review & editing:** Akira Ikeya, Mitsuko Nakashima, Miho Yamashita, Hirotomo
Saitsu, Yutaka Oki.

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
