## [Decision Letter · Decision Letter 0]

25 Feb 2020

PONE-D-20-01265

CCNB2 and AURKA overexpression may cause atypical mitosis in Japanese cortisol-producing adrenocortical carcinoma with TP53 somatic variant

PLOS ONE

Dear Dr. Ikeya,

Thank you for submitting your manuscript to PLOS ONE. After careful consideration, we feel that it has merit but does not fully meet PLOS ONE’s publication criteria as it currently stands. Therefore, we invite you to submit a revised version of the manuscript that addresses the points raised during the review process.

We would appreciate receiving your revised manuscript by Apr 10 2020 11:59PM. To enhance the reproducibility of your results, we recommend that if applicable you deposit your laboratory protocols in protocols.io, where a protocol can be assigned its own identifier (DOI) such that it can be cited independently in the future. For instructions see: http://journals.plos.org/plosone/s/submission-guidelines#loc-laboratory-protocols

We look forward to receiving your revised manuscript.

Kind regards,

Sumitra Deb, PhD

Academic Editor

PLOS ONE

Journal Requirements:

2. Please specify in your ethics statement whether participant consent was written or verbal. If verbal, please also specify: 1) whether the ethics committee approved the verbal consent procedure, 2) why written consent could not be obtained, and 3) how verbal consent was recorded.

Reviewers' comments:

Reviewer's Responses to Questions

**Comments to the Author**

1. Is the manuscript technically sound, and do the data support the conclusions?

Reviewer #1: Yes

2. Has the statistical analysis been performed appropriately and rigorously? 

Reviewer #1: Yes

3. Have the authors made all data underlying the findings in their manuscript fully available?

Reviewer #1: Yes

4. Is the manuscript presented in an intelligible fashion and written in standard English?

Reviewer #1: Yes

5. Review Comments to the Author

Reviewer #1: This is a nice study in which the authors have investigated molecular markers in adrenocortical markers that are of diagnostic and prognostic significance.

Table-1, 2, 3: some columns are missing in the current version. Please upload again.

Line 104: The authors should explain the rationale for interrogating only the selected genes in this study.

Line 293-299: I did not find supporting evidence in literature to attribute pathogenicity to PRKAR1A(c.545C>A:p.T182K) variant. In silico tools in by themselves are not sufficient to attribute pathogenicity to a variant. The reasons listed by the authors to consider this variant as pathogenic are insufficient and in the absence of functional evidence, this variant would best be considered as a “variant of uncertain significance”. The reference to the variant in Clinvar is for PRKAR1A:c.545C>T and not c.545C>A as reported here. The authors come to similar conclusion in the discussion, and hence it is not reasonable to list it as a pathogenic variant.

Also since ZNRF3 is an uncommon variant, the authors should consider discussing the clinical significance and role in pathogenicity in more detail.

Line 331: atypical mitotic figures were positive in all of the cases with TP53 somatic variants and Ki-67 index ≥ 10%. I would need to see the complete Table-2 for verification of this.

6. PLOS authors have the option to publish the peer review history of their article (what does this mean?). If published, this will include your full peer review and any attached files.

Reviewer #1: No

---

## [Author Response · Author response to Decision Letter 0]

27 Mar 2020

Response to Reviewer #1:

Comment:

Q1. Table-1, 2, 3: some columns are missing in the current version. Please upload again.

A1. I apologize for not being able to show the whole table in the original manuscript. In accordance with Reviewer #1 comment, the font size has been adjusted for Tables 1 and 3 so that all columns are visible in the revised manuscript. Table 2 has been changed to be shown as Figure 1 in the revised manuscript.

Q2. Line 104: The authors should explain the rationale for interrogating only the selected genes in this study.

A2. We sincerely accept comments made by Reviewer #1. In accordance with Reviewer #1 comment, we have added Line 106-119 in the revised manuscript explaining the reason why we selected the 12 genes analyzed as candidate genes this study. 

One of our objectives of this study was to investigate the factors for poor prognosis of adrenocortical carcinoma (ACC) and to consider the potential for treatment. First, we selected ten candidate genes of TP53-RB1 and Wnt pathways, to which TP53 and CTNNB1 belong. TP53 and CTNNB1 somatic variant was reported as a poor prognostic factor in the previous report. We speculated that somatic variants of genes involved in these two genes may also contribute to poor prognosis. Because these candidates ten genes variants were detailed in a previous paper, they were also suitable for comparative studies between Japanese and other races. We further selected PRKAR1A, since it is associated with cortisol production, which is a factor of poor prognosis. We also selected TERF2, which may be associated with disease progression due to shortened telomere length, among ACC driver genes.

Q3. Line 293-299: I did not find supporting evidence in literature to attribute pathogenicity to PRKAR1A c.545C>A (p.T182K) variant. In silico tools in by themselves are not sufficient to attribute pathogenicity to a variant. The reasons listed by the authors to consider this variant as pathogenic are insufficient and in the absence of functional evidence, this variant would best be considered as a “variant of uncertain significance”. The reference to the variant in Clinvar is for PRKAR1A:c.545C>T and not c.545C>A as reported here. The authors come to similar conclusion in the discussion, and hence it is not reasonable to list it as a pathogenic variant.

A3. We sincerely accept comments made by Reviewer #1.　As suggested, since PRKAR1A:c.545C>T and c.545C>A are not the same, the text describing references to variants of Clinvar has been deleted. Also, we have changed Table S5 because we agree it is correct to classify the PRKAR1A:c.545C>A (p.T182K) variant as "variant of uncertain significance".

Regarding to the variant of the PRKAR1A:c.545C>A (p.T182K), it is a de novo somatic variant, a very rare variant with no registration such as gnomAD, and its pathogenic significance is predicted by in silico prediction. The pathogenicity of some missense variants of the cAMP binding domain A, such as c.547G>T (p.A183Y), c.638C>A (p.213D), and c.438A>T (p.R146S) have been demonstrated in vitro, as shown in the new reference 41 and 42. The in silico prediction of those variants, which was proven to be pathogenetic in vitro, were ”Probably damaging” or ”Benign”. The PRKAR1A:c.545C>A (p.T182K) variant is likely to affect cAMP binding domain A and which will affect the enzyme function of PRKAR1A through conformational changes. It is presumed that the change enhances PKA activity and affects tumor formations. The PRKAR1A gene is one of the ACC driver genes, and variants in this gene are presumed to be clinically significant, especially in ACC. Based on the above, we have identified the PRKAR1A:c.545C>A (p.T182K) variant in this study as a potential variant of carcinogenesis.

We have added the above content to LINE 559-575 and references 41 and 42 as a new reference in the revised manuscript. 

Certainly, as pointed out by Reviewer #1, the effect on the actual carcinogenesis will need to be evaluated for the functional studies of the variant. We are aiming to demonstrate the actual functional evaluation of the PRKAR1A:c.545C>A (p.T182K) variant in the future.

Q4. Also since ZNRF3 is an uncommon variant, the authors should consider discussing the clinical significance and role in pathogenicity in more detail.

A4. In accordance with Reviewer #1 comment, we have added a detailed discussion of the clinical significance and role in the pathogenicity of ZNRF3: c.433C>T (p.R145X) variant at LINE 576-588 in the revised manuscript. The ZNRF3:c.433C>T (p.R145X) variant was a nonsense variant of the transmembrane domain that has been reported to have a high potential for affecting pathogenicity in previous reports. It is speculated that this variant may upregulate the Wnt pathways when variant position are considered. This variant is a very rare variant that is not registered in gnomAD and its pathogenic significance was predicted by in silico prediction. 

Q5. Line 331: atypical mitotic figures were positive in all of the cases with TP53 somatic variants and Ki-67 index ≥ 10%. I would need to see the complete Table-2 for verification of this.

A5. As indicated, in the revised manuscript, we changed Table2 to Figure1 so that you can check the contents of Table2 that Reviewer #1 pointed out.

---

## [Editor Report · Decision Letter 1]

30 Mar 2020

CCNB2 and AURKA overexpression may cause atypical mitosis in Japanese cortisol-producing adrenocortical carcinoma with TP53 somatic variant

PONE-D-20-01265R1

Dear Dr. Yamashita,

We are pleased to inform you that your manuscript has been judged scientifically suitable for publication and will be formally accepted for publication once it complies with all outstanding technical requirements.

With kind regards,

Sumitra Deb, PhD

Academic Editor

PLOS ONE
---

## [Editor Report · Acceptance letter]

1 Apr 2020

PONE-D-20-01265R1 

CCNB2 and AURKA overexpression may cause atypical mitosis in Japanese cortisol-producing adrenocortical carcinoma with TP53 somatic variant 

Dear Dr. Yamashita:

I am pleased to inform you that your manuscript has been deemed suitable for publication in PLOS ONE. Congratulations! Your manuscript is now with our production department. 

With kind regards,

on behalf of

Dr. Sumitra Deb 

Academic Editor

PLOS ONE